# Persistence and Variability of Ice Stream Grounding Lines on Retrograde Bed Slopes

Alexander A. Robel[1,3], Christian Schoof[2], and Eli Tziperman[1]

[1]Department of Earth and Planetary Sciences, Harvard University, Cambridge, Massachusetts, USA.
[2]Department of Earth and Ocean Sciences, University of British Columbia, Vancouver, British Columbia, Canada.
[3](Currently) Division of Geological and Planetary Sciences, California Institute of Technology, Pasadena, California, USA.

*Correspondence to:* Alexander A. Robel (robel@caltech.edu)

**Abstract.** In many ice streams, basal resistance varies in space and time due to the dynamically-evolving properties of subglacial till. These variations can cause internally-generated oscillations in ice stream flow. However, the potential for such variations in basal properties are not considered by conventional theories of grounding line stability on retrograde bed slopes, which assume that bed properties are static in time. Using a flowline model, we show how internally-generated, transient variations in ice stream state interact with retrograde bed slopes. In contrast to predictions from the theory of the marine ice sheet instability, our simulated grounding line is able to persist and reverse direction of migration on a retrograde bed when undergoing oscillations in the grounding line position. In turn, the presence of a retrograde bed may also suppress or reduce the amplitude of internal oscillations in ice stream state. We explore the physical mechanisms responsible for these behaviors and discuss the implications for observed grounding line migration in West Antarctica.

## 1   Introduction

Theoretical and numerical studies have shown that under certain conditions, the flux of ice through the grounding line (where ice transitions from grounded to floating) increases sensitively with bedrock depth (Weertman, 1974; Chugunov and Wilchinsky, 1996; Wilchinsky, 2001; Schoof, 2007a, b; Nowicki and Wingham, 2008; Durand et al., 2009; Robison et al., 2010; Fowler, 2011; Schoof and Hewitt, 2013). Thus, when the grounding line rests on a retrograde slope (sloping upwards in the direction of flow), any retreat in grounding line position leads to greater flux and consequently additional retreat, and any advance in grounding line position leads to decreased flux and additional advance. By preventing a grounding line on a retrograde or very shallow prograde bedrock slope from achieving a stable steady state position, this positive flux feedback causes the "marine ice sheet instability" (Weertman, 1974; Wilchinsky, 2009; Schoof, 2012).

Ice streams are regions of fast-flowing ice that account for over 90% of mass transport from the interior of the Antarctic Ice Sheet to the ocean (Bamber et al., 2000). Observations indicate that the flow of some ice streams exhibits unforced variability on centennial to millennial time scales that has a significant impact on the mass balance of the Antarctic Ice Sheet (Joughin and Tulaczyk, 2002). Other observed changes in the flow velocity of different West Antarctic ice streams have been attributed to ocean- and atmosphere-forced melting of ice shelves (e.g. Rignot et al., 2008; Pritchard et al., 2009), which buttress ice sheets through lateral contact with bedrock and localized basal contact with bathymetric high (Goldberg et al., 2009; Katz

and Worster, 2010; Gudmundsson et al., 2012). Such changes have raised the possibility that forced ice stream variability near topographic transitions to overdeepenings with sections of retrograde slope may lead to rapid retreat (e.g. Favier et al., 2014; Joughin et al., 2014). Studies have also begun considering additional or alternate physical processes that play a role in determining the location and stability of grounding lines in marine ice sheets, including gravitationally-driven changes in local sea level (Gomez et al., 2010, 2012), variations in ice stream trough width (Jamieson et al., 2012, 2014; Docquier et al., 2014), tidal compaction of till (Christianson et al., 2013) and sedimentation (Alley et al., 2007). Some of these processes have been shown potentially to stabilize grounding lines on retrograde slopes.

When interpreting observations of grounding line migration, a critical question is whether retreat on a retrograde slope necessarily implies continued irreversible retreat in the future due to the marine ice sheet instability. In this study, we explore how unforced, internal ice stream variability interacts with forced variability and the consequent limitations of interpreting observed grounding line retreat. Previous work has studied how stable and unstable steady state configurations for marine ice sheets with constant bed properties depend on forcing parameters such as accumulation rates (Schoof, 2007b), showing that irreversible transitions across overdeepenings happen at critical values of these forcing parameters. Similarly, for ice streams on purely downward-sloping beds but with thermomechanically evolving bed properties, other work has shown that steady ice streams can undergo bifurcations into oscillatory behavior when the same forcing parameters are changed (e.g. Robel et al., 2013, 2014). Here we combine the two approaches and ask how changes in accumulation rate and other parameters change both, the stable steady states and the stably oscillating configurations of an ice stream model with evolving bed properties and an overdeepening. A much richer set of transitions (or bifurcations) is possible: instead of simple jumping from steady states on either side of the overdeepening, these steady states can also transition into oscillatory states, and these oscillatory states can potentially encroach into the overdeepening, or transition to steady states on the other side of the overdeepening.

This study uses a flowline model that incorporates dynamically-varying till properties, including temperature and water content. The ice stream has constant width, geothermal heat flux and surface temperature in both space and time. We do not include buttressing in our simulations in order to focus our analysis on time-dependent variations in bed properties, and exclude the potential influence of an ice shelf on grounding line stability (as discussed extensively in previous studies, such as Goldberg et al., 2009; Katz and Worster, 2010; Gudmundsson et al., 2012), though we do discuss the importance of buttressing further in section 5. We examine the processes which play a role in unforced, coupled oscillations in bed properties, ice stream flow and grounding line position on and near retrograde slopes. We come to two main conclusions. First, allowing bed properties to dynamically evolve radically changes model predictions of grounding line position and variability. Second, ice streams, like those in the Siple Coast region of West Antarctica, can exhibit behavior that is unexplained by existing theories of the marine ice sheet instability and ice stream variability. This includes persistence of the grounding line on a retrograde slope for centuries in the absence of buttressing and a reversal of migration direction on the retrograde slope. Other notable examples of the effect of retrograde slopes include the reduction or suppression of unforced oscillations in the state of an unbuttressed ice stream in ways that are not predicted by theories of ice stream variability which do not consider complex topography. We conclude that it is important to consider these diverse dynamical responses to forcing and bed topography when evaluating whether observed

grounding line retreat onto a retrograde slope is irreversible (Favier et al., 2014; Joughin et al., 2014) and in predicting future ice sheet change.

In section 2, we describe the ice stream flowline model, idealized bed configuration and underlying mechanism of unforced ice stream variability. In section 3, we analyze the dependence of ice stream grounding line variability on accumulation rate and initial position for a set bed topography. We also compare the behavior of the ice stream when bed properties are not permitted to vary and when they vary freely. In section 4, we systematically vary the idealized topographic configuration and discuss the physical mechanisms which cause modification of ice stream variability without a retrograde section. Specifically, we analyze how the position, length and slope of the retrograde bed section lead to different types of ice stream behavior. In section 5, we discuss the relevance and implications of these findings for more complex models and ice stream observations.

## 2 Model preliminaries

Active ice streams are strongly resisted by lateral shear stresses resulting from contact with topography or slow-flowing ice ridges (Echelmeyer et al., 1994). Additionally, many ice streams are underlain by till which behaves like a Coulomb plastic material with yield stress depending on water content (Tulaczyk et al., 2000a). In this study, we employ a shallow ice stream flowline model which includes lateral shear stresses and dynamically evolving subglacial till properties. This model solves for ice thickness and till water content along a central flowline ($x$), with horizontal velocity, vertical velocity and ice temperature also resolved in the vertical ($z$). The horizontal force balance is

$$\frac{\partial}{\partial x}\left(2h\bar{A}^{-\frac{1}{n}}\left|\frac{\partial u_b}{\partial x}\right|^{\frac{1}{n}-1}\frac{\partial u_b}{\partial x}\right) = \rho_i gh\frac{\partial h}{\partial x} + \tau_b + G_s h|u_b|^{\frac{1}{n}-1}u_b,$$ (1)

incorporating integrated lateral shear stress and basal shear stress from an undrained Coulomb plastic bed that evolves as meltwater is produced and refreezes at the ice-till interface. $\bar{A}$ is the depth-averaged Nye-Glen Law coefficient which is a function of ice temperature, $n$ is the Nye-Glen Law exponent, $\rho_i$ is the density of glacier ice, $g$ is the acceleration due to gravity and $G_s \propto W^{-1}$ is a parameter capturing the importance of lateral shear stress where $W$ is the ice stream half-width. At the grounding line, longitudinal stress is balanced by water pressure (Shumskiy and Krass, 1976)

$$\left[2\bar{A}^{-\frac{1}{n}}h\left|\frac{\partial u_b}{\partial x}\right|^{\frac{1}{n}-1}\frac{\partial u_b}{\partial x}\right]_{x=x_g} = \frac{1}{2}\rho_i g\left(1-\frac{\rho_i}{\rho_w}\right)h(x_g)^2.$$ (2)

The upstream boundary is defined to be the ice divide and correspondingly, velocity is set to zero there: $u_b(x=0)=0$.

Vertical shear of horizontal velocity is a function of driving stress

$$u(z) = u_b + \frac{2\bar{A}}{n+1}\tau_d^n h\left[1-\left(1-\frac{z-b}{h}\right)^{n+1}\right],$$ (3)

| Parameter | Description | Value |
|---|---|---|
| $b_0$ | Ice divide bed height (m) | 100 |
| $b_x$ | Prograde bed slope | $5 \times 10^{-4}$ |
| $g$ | Acceleration due to gravity ($\mathrm{m \cdot s^{-2}}$) | 9.81 |
| $G$ | Geothermal heat flux ($\mathrm{W \cdot m^{-2}}$) | 0.07 |
| $G_s$ | Lateral shear stress parameter ($\mathrm{kg \cdot s^{-4/3} \cdot m^{-7/3}}$) | 400 |
| $n$ | Nye-Glen Law exponent | 3 |
| $T_s$ | Ice surface temperature ($^{\circ}$C) | -15 |
| $Z_0$ | Maximum available till thickness (m) | 4 |
| $\rho_i$ | Ice density ($\mathrm{kg \cdot m^{-3}}$) | 917 |
| $\rho_w$ | Seawater density ($\mathrm{kg \cdot m^{-3}}$) | 1028 |

**Table 1.** Parameters used in this study (unless varied as indicated in Table 2). Other parameters used in the model, but not discussed in the text are listed in Robel et al. (2014).

and is added to the basal velocity calculated from equation 1 to form the full velocity field $u(x, z)$. Vertical velocity is determined by integrating the mass continuity equation upwards from the bed at constant $x = x_0$

$$w(x, z, t) = - \int_{b}^{b+h} \frac{\partial u}{\partial x}\bigg|_{x=x_0} dz, \tag{4}$$

subject to the condition $w + u\frac{\partial b}{\partial x} = 0$ at $z = b$, where basal melt rate is neglected as it is typically less than an order of magnitude

smaller than vertical velocity in model simulations.

The evolution of ice thickness follows a mass conservation equation with a source term due to accumulation

$$\frac{\partial h}{\partial t} + \frac{\partial}{\partial x}(\bar{u}h) = a_c, \tag{5}$$

where $\bar{u}$ is the depth-averaged horizontal velocity and $a_c$ is the accumulation rate. At the grounding line, ice is at the flotation thickness

$\rho_i h(x_g) = \rho_w b(x_g), \tag{6}$

where $\rho_i$ is the density of glacial ice and $\rho_w$ is the density of seawater. Ice flux through the grounding line ($q_g = u_b(x_g)h(x_g)$) is removed from the grounded ice stream system. When the bed slope is locally non-zero at the grounding line, accumulation and local imbalances of advective ice flux cause changes in ice thickness which require migration of the grounding line to maintain this flotation condition.

Basal heat budget determines the basal melt rate

$$m = \frac{1}{\rho_i L_f}\left(G + \tau_b u_b + k_i \frac{\partial T}{\partial z}\bigg|_{z=b}\right), \tag{7}$$

where, on the right hand side, the first term is the geothermal heat flux, the second term is the frictional heat flux and the third term is the vertical conductive heat flux at the bed. $k_i$ is the thermal conductivity of glacial ice and $L_f$ is the latent heat of fusion. Till water content ($w = eZ_s$ where $e$ is the till void ratio and $Z_s$ is the thickness of unfrozen subglacial till) changes as

basal heating produces or freezes meltwater

$$\frac{\partial (e Z_s)}{\partial t} = m, \tag{8}$$

Till behaves as a Coulomb plastic material, and so basal shear stress is calculated from the basal velocity and void ratio

$$\tau_b = \tau_c \frac{u_b}{\sqrt{u_b^2 + \epsilon_u^2}}, \tag{9}$$

where $\epsilon_u$ is the velocity scale over which till transitions from a quasi-linear to Coulomb friction law. The critical failure strength of the till follows the empirical form of Tulaczyk et al. (2000a)

$$\tau_c = \tau_0 \exp[-b(e - e_c)], \tag{10}$$

where $\tau_0$ and $b$ are empirical parameters.

Ice temperature is modeled by an advection-diffusion equation in the $x - z$ plane

$$\frac{\partial T}{\partial t} + \frac{\partial}{\partial x}(uT) + \frac{\partial}{\partial z}(wT) = \kappa \left( \frac{\partial T}{\partial x^2} + \frac{\partial T}{\partial z^2} \right), \tag{11}$$

where $\kappa$ is the thermal diffusivity of glacial ice. We neglect strain heating within the ice which is negligible at the central ice stream flowline, though it may be important under specific circumstances within shear margins (see discussion in section 5). At the ice surface, the temperature is equal to a prescribed atmospheric temperature: $T(z = b + h) = T_s$.

The model accurately simulates transient migrations in grounding line position and activation waves, propagating fronts associated with the transition from a strong, non-deforming bed to a weak, deforming bed during ice stream activation (Fowler and Schiavi, 1998). The simulations in this study are run at very high horizontal spatial resolution in the grounding zone (~100 m) and high resolution elsewhere (~1 km). At these horizontal resolutions, the range of simulated grounding line migration is converged and so small changes in the mesh size do not significantly change the results discussed. The vertical resolution is sufficient (~10 m) to resolve the vertical temperature gradient of basal ice. Additional details of the flowline model and the numerical approach used in the following simulations can be found in Robel et al. (2014).

In certain parameter regimes, the ice stream grounding line migrates as a part of internal thermal oscillations in the absence of a retrograde section on the bed (Clarke, 1976; Fowler et al., 2001; Sayag and Tziperman, 2009, 2011; Robel et al., 2013). These thermal oscillations (also known as "binge-purge" oscillations) are due to a similar physical mechanism as thermal surging in mountain glaciers, described elsewhere (e.g. Robin, 1955; Oerlemans, 1982; Fowler, 1987; MacAyeal, 1993). A typical oscillation proceeds as follows: when the ice stream activates, till is weak and ice stream horizontal velocity is high. As ice is advected from upstream, the grounding line thickens and rapidly advances by $100 - 150$ km to its most seaward position. Ice in the active ice stream trunk thins due to an "overshoot" of high advective ice flux which exceeds accumulation, leading to increased vertical heat conduction, freezing basal meltwater and strengthening the bed. During the second part of the active phase, ice stream velocity begins to decrease and the grounding line retreats. Eventually, till becomes sufficiently strong that the combined basal shear strength and lateral shear stress exceeds the driving stress and the ice stream stagnates. During the stagnant phase, the grounding line continues to retreat slowly to its minimum grounding line position. The ice stream slowly

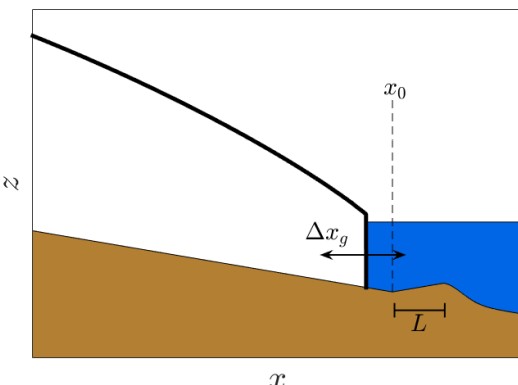

**Figure 1.** Schematic of bed configuration for model experiments. Solid black line indicates ice sheet profile. Brown shaded region is bedrock. Blue shaded region is seawater. $x_0$ is the horizontal position where the retrograde section begins. $L$ is the length of the retrograde section. $\Delta x_g$ indicates the range over which the grounding line migrates during thermal oscillations.

| Figure | Vary bed | $a_c$ (m/yr) | $x_0$ (km) | $L$ (km) | $b_{xr}$ |
|--------|----------|--------------|------------|----------|----------|
| 2a | No | $0.05 - 0.8$ | 800 | 80 | $2 \times 10^{-4}$ |
| 2b | Yes | $0.05 - 0.8$ | 800 | 80 | $2 \times 10^{-4}$ |
| 2c | Yes | $0.05 - 0.8$ | 800 | 80 | $1 \times 10^{-3}$ |
| 2d | Yes | $0.05 - 0.8$ | – | – | – |
| 3a-d (dash) | Yes | 0.3 | – | – | – |
| 3a (solid) | Yes | 0.3 | 700 | 40 | $2 \times 10^{-4}$ |
| 3b (solid) | Yes | 0.3 | 800 | 80 | $2 \times 10^{-4}$ |
| 3c (solid) | Yes | 0.3 | 700 | 140 | $6 \times 10^{-4}$ |
| 3d (solid) | Yes | 0.3 | 700 | 140 | $8 \times 10^{-4}$ |
| 4 (solid) | No | 0.32 | 800 | 80 | $2 \times 10^{-4}$ |
| 4 (dash) | No | 0.59 | 800 | 80 | $2 \times 10^{-4}$ |
| 5a-b | Yes | 0.3 | 800 | 0-200 | $1 - 10 \times 10^{-4}$ |
| 5c-d | Yes | 0.3 | 800 | 0-200 | $1 - 10 \times 10^{-4}$ |

**Table 2.** Prescribed parameters and their variation in different simulations listed by figure. Dashed entries indicates no retrograde section. A "yes" in the vary bed column indicates simulations in which bed properties are permitted to vary as a function of till water content.

thickens upstream, which reduces the advection of cold ice to the bed and warms basal ice through conduction. Basal meltwater is produced and eventually weakens the bed sufficiently that activation occurs. The dynamics of this process are discussed in further detail in Robel et al. (2014).

To explore how internal ice stream variability interacts with retrograde bed topography typically associated with overdeep-
5 enings and sediment wedges, we add a section of retrograde slope to prograde topography (schematically illustrated in Figure 1). The resulting bed topography is given by

$$b(x) = b_0 - b_x x + b_r(x). \tag{12}$$

where $b_0$ is the bed elevation at the ice divide and $b_x = 5 \times 10^{-4}$ is the background prograde slope. The added topography, $b_r(x)$, has a section of linear retrograde slope, which then exponentially relaxes back to the prograde profile

$$b_r(x) = \begin{cases} 0 & \text{if } x < x_0 \\ b_{xr}(x - x_0) & \text{if } x_0 \le x \le x_0 + L \\ L b_{xr} \exp\left[ -\left( \frac{2(x - x_0 - L)}{L} \right)^2 \right] & x > x_0 + L \end{cases}, \tag{13}$$

where $x_0$ is the position of the beginning of the retrograde section, $L$ is the length of the retrograde section and $b_{xr}$ is the slope of the retrograde section. We choose this simplified bed topography so that we can systematically vary $x_0$, $L$ and $b_{xr}$ over a wide range of reasonable bed configurations (as indicated in Table 2).

## 3    Spatiotemporal variations in bed properties change grounding line behavior

The thermally-induced oscillations described in section 2 rely on till water content and yield strength evolving in response to net basal heat flux. By contrast, the marine ice sheet instability paradigm of steady state grounding lines jumping across overdeepenings (Schoof, 2007a; Pattyn et al., 2012) is based on bed properties (here, the basal yield stress) that remain constant in time, as is frequently assumed in ice stream simulations. With the bed configuration defined in equations 12 and 13, we can reproduce the same behavior if we set the basal yield stress to a constant value, chosen to be zero for simplicity. We systematically vary the accumulation rate $a_c$ as a forcing parameter, and run the model to steady state for each value, with the fixed basal yield stress ensuring that no oscillations occur. Figure 2a shows the resulting steady state grounding line position on the horizontal axis as a function of accumulation rate plotted on the vertical axis. The bed geometry parameters used here are $x_0 = 800$ km, $L = 80$ km, with $b_{xr} = 2 \times 10^{-4}$. By starting the simulations with initial grounding line positions on either side of the retrograde section, we are able to reproduce the expected hysteresis, analogous to, for example, Figure 9a of Schoof (2007b). For a range of accumulation rates $0.35 \lesssim a_c \lesssim 0.6$ m/yr, multiple stable steady state solutions are possible, and critical values of $a_c$ exist, at which the large and small ice sheet solution branches disappear when the grounding line moves onto the retrograde slope. In this region, grounding line advance leads to further advance due to increased accumulation and decreasing mass loss on the shallowing bed, while grounding line retreat leads to further retreat due to decreased accumulation and increased mass loss on the deepening bed.

There are some subtle differences between our model and that used in Schoof (2007a) or the MISMIP hysteresis experiments in Pattyn et al. (2012); instead of a Weertman-type basal friction law $\tau_b = C|u|^{m-1}u$, the solutions in Figure 2a have $\tau_b = 0$ and the dominant drag term is the lateral drag $G_s h |u_b|^{\frac{1}{n}-1} u_b$ (see also Jamieson et al., 2012). However, as Hindmarsh (2012) demonstrates, the relationship between flux and ice thickness at the grounding line that underpins the hysteresis in Schoof (2007a) and Pattyn et al. (2012) carries over to the case of lateral drag dominated flow with a simple change in coefficients and exponents.

In Figure 2b-d, till water content and basal yield stress are allowed to dynamically evolve (with other prescribed constants as in Tables 1 and 2). Again, the model is run forward for many different values of accumulation rate $a_c$ until, for each $a_c$, the solution settles into either a limit cycle or steady state. Note that aperiodic oscillations of the type identified in Brinkerhoff and

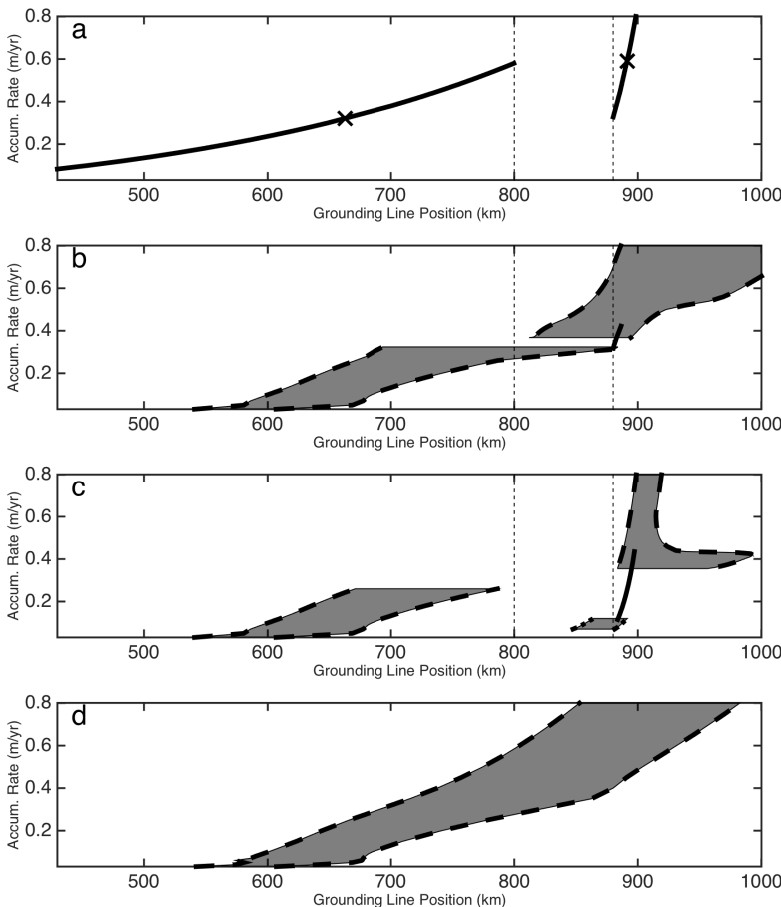

**Figure 2.** Stable equilibrium grounding line positions and limit cycle grounding line oscillation amplitudes as function of accumulation rate $a_c$. Solid lines indicate stable equilibrium grounding line positions. Shading bounded by dashed lines indicates range over which grounding line oscillates. (a) Bed properties are static with $\tau_c = 0$ kPa everywhere. $b_{xr} = 2 \times 10^{-4}$ (b) Bed properties dynamically evolve. $b_{xr} = 2 \times 10^{-4}$. (c) Bed properties dynamically evolve. $b_{xr} = 10^{-3}$. (d) Bed properties dynamically evolve with no retrograde section. In panels a-c $x_0 = 800$ km and $L = 80$ km. "x" marks indicate location of steady states for simulations plotted in Figure 4.

Johnson (2015) do not appear in our flowline solutions, likely due to the lack of interactions with other ice streams. For many values of the accumulation rate, we see limit cycle solutions, and the range of grounding line migration for each of these limit cycles is indicated by the shaded regions in Figure 2b-d. Stable steady states are indicated by solid lines.

The abrupt transitions from steady states on one side of the retrograde section to the other (seen in Figure 2a) no longer

5   appear when bed properties are allowed to evolve for the same bed topography (Figure 2b). Instead, we observe that, for small

accumulation rates $a_c$, we have only limit cycles whose amplitude increases with $a_c$, but no steady states. For small enough $a_c$, the grounding line remains on the landward side of the retrograde section. For larger accumulation rates we see the limit cycles eventually encroach on the retrograde section, and eventually the grounding line migrates all the way across the retrograde section during a single limit cycle. At a critical value of $a_c \approx 0.37$ m/yr, the limit cycle abruptly disappears, and is replaced by

5 a stable steady grounding line on the far side of the retrograde section. A further increase in accumulation rate then destabilizes the steady state, and leads to renewed oscillations that see the grounding line move between the retrograde section and the seaward side, with the range of oscillation moving gradually away from the retrograde section for even larger $a_c$. Note that there are two instances of hysteresis here, too. In this case, however, the hysteresis reflects the possibility of the ice sheet settling either into a steady state or into a limit cycle for a small range of values of $a_c$ between $0.317$ and $0.323$ m/yr and

10 between $0.37$ and $0.42$ m/yr. That is fundamentally different from the hysteresis in Figure 2a, with the possibility of the ice sheet settling into two or more different steady states.

Using our simple approach, we cannot properly characterize the bifurcations that mark the appearance or disappearance of a steady state, and our method may also have difficulty in capturing solutions close to the bifurcations. For the smoother bed and slightly different physics in Schoof (2007a), the bifurcations that mark the appearance or disappearance of the two steady

state solution branches in Figure 2a can be identified as saddle-node bifurcations. The much simpler box model for thermally-induced oscillations in Robel et al. (2013) would identify the hysteresis associated with the disappearance of the steady state in favor of a limit cycle as a combination of a subcritical Hopf bifurcation and a saddle-node bifurcation of limit cycles. Whether those bifurcations remain relevant to the present spatially extended (that is, high-dimensional) dynamical system is unclear.

In Figure 2c, we recompute steady states and limit cycle solutions for the same parameter values as in Figure 2b, but with

20 the retrograde section slope steepened to $b_{xr} = 10^{-3}$. Here we observe a much clearer separation between steady states and limit cycles that remain on one side or the other of the retrograde section. For small $a_c$, we have only oscillatory solutions in which the grounding line remains on the landward side of the retrograde section, and for large $a_c$, we have oscillatory solutions where the grounding line remains on the seaward side. The solution structure is now considerably more complicated, however, with stable equilibria existing on the seaward side of the retrograde section for a larger range of $a_c$, and yet more hysteresis

permitting the coexistence of that steady state with either one or two limit cycle solutions. Notably, the limit cycle on the landward side for small $a_c$ never encroaches on the retrograde section, but there is a limit cycle centered on the seaward side for $0.12 \lesssim a_c \lesssim 0.2$ m/yr, during which the grounding line periodically retreats into the retrograde section, only to readvance out of it again. Figures 2b-c should also be contrasted with Figure 2d. Here we recompute solutions for the same parameter values but with no retrograde section ($L = 0$ km), demonstrating that the more complex dynamics and hysteresis apparent in

Figures 2b-c are intrinsically related to the presence of a retrograde section, without which the same parameter values invariably lead to oscillations.

Some solution branches found through numerical integration of the model in this study occupy a narrow region of parameter space. It is unclear whether real ice streams would ever occupy this part of parameter space given that natural variability and measurement errors in accumulation rate and other environmental variables are typically larger than the range associated

with these solutions. Notwithstanding these narrow solution branches, the large bifurcations that separate most of the different

grounding line solutions indicate that there are robust differences in ice stream behavior over realistic parameter ranges that are readily observable in real ice streams. The presence of narrow regimes of ice stream behavior emphasize the need for perturbed-physics ensemble studies (Murphy et al., 2004) with more complex ice sheet models to ensure that simulated ice sheet behaviors are broadly representative of the full range of possibilities.

Comparing the static and dynamic bed cases in Figure 2, it is clear that allowing spatiotemporal variations in bed properties changes not just the grounding line position, but the behavior of the grounding line, over a wide range of possible values of accumulation rate ($a_c$). In some parameter regimes, thermal oscillations cause the grounding line to migrate onto the retrograde section (during ice stream stagnation or activation) before reversing direction. Such behavior cannot be explained by appealing to the flux feedback which causes the marine ice sheet instability for a static bed, as in Figure 2a. That flux feedback relies on

a relationship between grounding line flux and ice thickness. For sliding at the ice sheet bed, relationships of this kind can be justified by boundary layer theories (Schoof, 2007a, b; Tsai et al., 2015), and involve prescribed physical properties of the bed such as a friction coefficient or yield stress. Unstoppable retreat or advance on retrograde slopes then occurs if these properties do not evolve as the ice sheet geometry does. If, for instance, the grounding line retreats onto the retrograde slope but the basal yield stress increases after it does so, there is no reason why the grounding line could not readvance subsequently. We consider

the relevant physical processes that permit such behavior in more detail in the next section.

## 4    Ice stream behavior not explained by simple flux-thickness feedbacks

The goal of this section is to explore the range of ice stream behaviors that are caused by the interaction of thermomechanical feedbacks with geometrical effects for different retrograde bed topographies and explain the physical mechanisms which cause some of these behaviors to diverge from the predictions of earlier theories. To do so, the accumulation rate is initialized

at $a_c = 0.1$ m/yr (with grounding line position at approximately $640$ km) and then slowly increased to $a_c = 0.3$ m/yr. As discussed in section 2, buttressing stress at the grounding line is not included in this study. In a baseline simulation, there is no retrograde bed topography, and the dynamically evolving bed causes the grounding line position to oscillate (as described in section 2) between $690$ km and $815$ km. The simplified nature of the bed topography (equations 12-13) then permits the addition of a section of retrograde bed that may modify the baseline oscillatory behavior. In each experiment, the position

($x_0$), length ($L$) or slope ($b_{xr}$) of the retrograde section can be changed (Table 2), while all other parameters are held constant (Table 1). Changing these geometrical parameters controlling bed topography explores a section of parameter space that is orthogonal to that explored in section 3 (where accumulation was varied). By comparing the baseline simulation to ice stream behavior with an added section of retrograde slope, we can then explain how natural modes of ice stream variability interact with bed topography. In an exploration of the parameter space of potential retrograde bed configurations, we find four types

of ice stream behavior. Figure 3 shows (after a period of transient initialization) four representative simulations of grounding line migration (solid lines) with the extent of the retrograde section in these simulations shaded in grey. For comparison, the baseline simulation is plotted as a dashed line in all panels. This includes ice stream grounding lines which exhibit: (a) persistence on the retrograde section for centuries during ice stream stagnation before reversing direction of migration; (b)

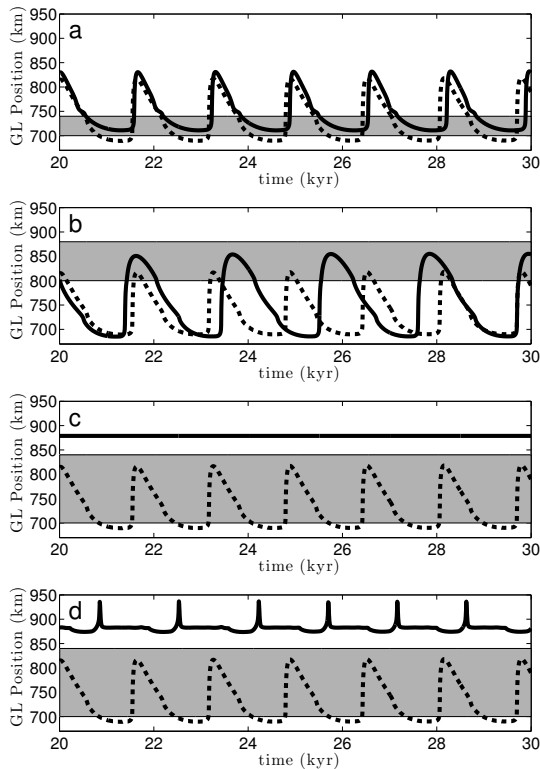

**Figure 3.** Four representative examples of interaction between retrograde section and ice stream thermal oscillations. In all panels, solid line is simulated grounding line migration after transient initialization period ($t < 20$ kyr), dashed line is grounding line migration in baseline simulation run without any retrograde section ($L = 0$ km) and grey shaded area is extent of retrograde section. (a) Minimally-modified thermal oscillations with ice stream persistence on the retrograde section during stagnation ($b_{xr} = 2 \times 10^{-4}$). (b) Amplified thermal oscillations with part of active phase on retrograde section ($b_{xr} = 2 \times 10^{-4}$). (c) Suppressed thermal oscillations ($b_{xr} = 6 \times 10^{-4}$). (d) Thermal oscillations with reduced amplitude ($b_{xr} = 8 \times 10^{-4}$). All examples are initialized with $x_g = 640$ km.

amplified variability and reversal of direction of migration while active on the retrograde section; (c) complete suppression of variability; (d) reduced amplitude of variability.

## 4.1 Persistence of grounding line on retrograde slope during ice stream stagnation

When the retrograde section is either short or located far upstream of the grounding line, oscillations in grounding line position
5  (Figure 3a) are similar in amplitude and period to the baseline simulation (dashed line), though offset slightly in position along

the bed. In such cases, the grounding line does not have much (or any) distance over which it interacts with the retrograde section, thus minimizing the departure from the baseline simulation where there is no retrograde section.

In a subset of cases where the retrograde section is located around the minimum position of the grounding line from the baseline simulation (including the simulation in Figure 3a and where $0.43 \lesssim a_c \lesssim 0.7$ m/yr in Figure 2b), the grounding line retreats onto the retrograde section, remains there for the duration of the stagnant phase (hundreds to thousands of years) and then reverses direction and advances onto the prograde slope. We can understand the mechanism of this behavior by comparing with the static bed case. The solid line of Figure 4 shows the transient evolution of a single simulation with static bed properties (where the final steady state is marked by an "x" on Figure 2a), initialized at $x = 881$ km and then slowly forced to retreat onto the seaward edge of the retrograde section (at $x_0 + L = 880$ km). As we would expect, the marine ice sheet instability causes irreversible retreat over 1500 years due mass loss from increasing flux through the grounding line. In contrast, when bed properties are allowed to freely vary (Figure 3a), such an irreversible retreat may not occur because yield stress of the bed is increasing and the intuition derived from the static bed theory of the marine ice sheet instability does not hold. In the second half of the active phase and into the beginning of the stagnant phase, grounding line retreat is the result of thinning of the grounding line, which is itself caused by flux through the grounding line exceeding the supply of ice advected from upstream. Following stagnation, however, the rapidly strengthening bed significantly reduces the mass flux out of the ice stream, slowing the rate of grounding line migration that would otherwise manifest as irreversible retreat across the entire retrograde slope (as in the solid line of Figure 4). During this slow retreat, a reservoir of excess ice mass accumulates upstream, driving an increasing gradient in ice thickness and driving stress, which eventually exceeds the yield strength of partially frozen till and causes slow sliding in the grounding zone. This slow sliding leads to initial ice thinning at the grounding line, reversal of the direction of migration and advance back onto the downstream prograde slope. At this point enough meltwater is produced through frictional heating (and insulation provided by the thick ice) that the ice stream activates further upstream, delivering significant mass to the grounding line, leading to thickening and rapid advance.

## 4.2 Active ice stream grounding line reversing direction of migration on a retrograde slope

When there is a shallow retrograde section located around the maximum grounding line position (Figure 3b), the active ice stream advances onto the retrograde section, persists for a few centuries, then reverses direction and retreats to the prograde slope. The oscillations in ice stream grounding line position are also amplified from the baseline simulation with no retrograde section (also where $0.27 \lesssim a_c \lesssim 0.32$ m/yr in Figure 2b).

To explain the mechanism behind this persistence on the retrograde section, we again draw a comparison with the static bed case from section 3. In a transient simulation (dashed line Figure 4), an ice stream with static bed properties is initiated at $x = 799$ km and gently forced onto the landward edge of the retrograde section (at $x_0 = 800$ km). As flux through the grounding line decreases, the grounding line thickens and advances over the course of 150 years until reaching prograde bed topography. In contrast, when bed properties are allowed to freely vary, high ice velocity during the initial part of the active phase delivers significant ice flux from upstream to the grounding line, leading to thickening and advance. Simultaneously, the flux from the upstream portion of the ice stream is too high to maintain a steady-state, leading to an "overshoot" where the

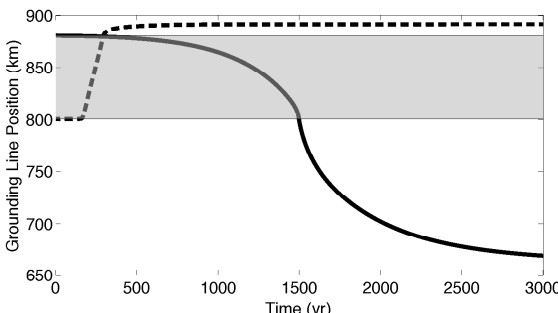

**Figure 4.** Transient evolution of ice stream grounding line position with bed properties kept static ($\tau_c = 0$ everywhere). $x_0 = 800$ km and $L = 80$ km and $b_{xr} = 2 \times 10^{-4}$. Solid line is initialized from $x_0 = 881$ km and $a_c = 0.33$ m/yr, decreasing to 0.32 m/yr over the first 300 years. Dashed line is initialized from $x_0 = 799$ km and $a_c = 0.58$ m/yr, increasing to 0.59 m/yr over the first 300 years.

upstream portion of the ice stream thins, leading to increased vertical heat conduction and till strengthening. Over a matter of decades in the latter part of the active phase, this till strengthening leads to stagnating velocities starting as a deactivation wave (Fowler and Schiavi, 1998; Robel et al., 2014) at the ice divide. As the grounding line advances onto the retrograde section, the upstream reservoir of ice is exhausted and thinning occurs at the grounding line as grounding line flux exceeds supply

from upstream. Then, when the deactivation wave reaches the grounding line, flux through the grounding line shuts down and ice at the grounding line begins to thicken, which briefly wins out over unstable advance over the retrograde slope and leads to a reversal in the direction of grounding line migration. The retrograde section is important, as it causes the grounding line to advance further than it would in the absence of a retrograde section, effectively amplifying thermal oscillations. If the retrograde section is steepened, there is an acceleration in speed of unstable grounding line advance related to the marine ice

sheet instability, which can overcome changes in ice flow due to thermal oscillations (see discussion in section 4.3).

    The case illustrated in this section is qualitatively similar to the persistence of a stagnant ice stream grounding line on a retrograde section discussed in section 4.1. The only difference here is that the retrograde section causes a larger advance during activation than would be expected from thermal oscillations alone (in the baseline experiment with no prograde slope). Nonetheless, the grounding line of any ice stream undergoing thermal oscillations will repeatedly reverse direction, even in

the absence of a retrograde slope (see dashed line in Figure 3b for baseline simulation). When the ice stream has built up or depleted a large reservoir of ice and is furthest from balancing mass accumulation and grounding line discharge, a change in ice flow may also cause thickening or thinning at the grounding line and thus change the direction of migration. In the cases highlighted in Figure 3a,b the retrograde sections do not change the ice stream behavior qualitatively, but may change the amplitude and period of thermal oscillations. In the following two sections we highlight cases where retrograde topography

changes the dynamics of the ice stream in a fundamental, qualitative way.

### 4.3 Suppression of thermal oscillations by a retrograde slope

A steep retrograde section located around the maximum grounding line position of the baseline simulation can completely suppress thermal oscillations (Figure 3c), leading to a steady-streaming state with no temporal variability in ice flow (Tulaczyk et al., 2000b; Sayag and Tziperman, 2009, 2011) and a grounding line positioned off the retrograde slope.

To explain the suppression of ice stream oscillations, we can examine how the retrograde slope interacts with the competing feedbacks that cause thermal oscillations. When the ice stream activates, there is an internal positive feedback between frictional heating, meltwater production and till weakening. There is also an internal negative feedback, where the grounding line advances, the ice stream thins and the rate of vertical heat conduction at the bed increases. In the baseline simulation without the retrograde section, the positive feedback initially dominates, causing the "overshoot" in horizontal velocity, ice thinning, significant heat loss due to vertical heat conduction, meltwater freezing and eventually stagnation. By introducing a retrograde section that is sufficiently long or steep, we increase the extent of grounding line advance during activation and the associated internal negative feedback, which counteracts the internal positive feedback of frictional heating. The result is that till weakens less during activation than in the baseline simulation, the ice stream achieves lower peak horizontal velocity and does not overshoot. Vertical heat conduction at the bed is compensated by geothermal heat flux and frictional heating, causing the ice stream to reach a steady state where the grounding line position is advanced further than the retrograde section.

Alternatively, suppression of thermal oscillations by a sufficiently long or steep retrograde section can be explained in terms of the stability criterion for thermal oscillations derived in Robel et al. (2013), which predicts that (everything else being equal) a longer ice stream enhances the importance of a constant geothermal heat flux relative to fluctuations in vertical conductive heat loss. By forcing the grounding line to advance, the retrograde section diminishes the amplitude of variations in vertical conductive heat loss, making it more likely that the basal heat budget will come into balance and the ice stream will reach a steady-streaming state.

### 4.4 Reduction of thermal oscillation amplitude by a retrograde slope

When the retrograde section is both long and steep, the amplitude of grounding line oscillations is reduced from the baseline simulation with no retrograde section (Figure 3d). These reduced-amplitude oscillations (which also occur where $0.5 \lesssim a_c \lesssim 0.8$ m/yr in Figure 2c) are completely limited to a range beyond the seaward end of the retrograde section ($x_0 + L$).

We can explain the reduction in thermal oscillations amplitude by starting from the suppressed state described in section 4.3. The long, steep retrograde slope forces the ice stream to advance and eventually (after transient evolution) settle down to a steady-streaming state similar to the behavior of suppressed oscillations. The difference here is that the retrograde section is sufficiently high ($> 100$ m) due to its steepness that the ice thickness above that retrograde slope is correspondingly thin (compared to the baseline scenario). Thus, it is not the fact that there is a retrograde section per se, but rather that this section of the bed is raised relative to the background prograde topography. Over the retrograde section, this leads to anomalously high vertical conductive heat loss from the bed, freezing of meltwater and strengthening of till. As till strengthens, velocity decreases and longitudinal stresses rapidly spread this signal upstream and downstream to the grounding line as a deactivation

wave (Fowler and Schiavi, 1998; Robel et al., 2014). Subsequently, accumulation causes thickening of stagnant ice and leads to reactivation, within a span of 100-200 years. This behavior is the same as typical thermal oscillations, with the only difference being that fluctuations in till state are restricted to a short portion of the bed (~200 km in this example) mostly downstream of the retrograde slope. During the remainder of the ~1000 years of the reduced thermal oscillation cycle, the ice stream is near the same balance as in the suppressed regime and the grounding line is nearly stationary.

## 4.5    A parameter space picture of retrograde section modification of ice stream oscillations

The four panels of Figure 5 map thermal oscillation amplitude as a function of retrograde section length ($L$) and bed slope ($b_{xr}$), for $x_0 = 700$ km (panels a,b) and $x_0 = 800$ km (panels c,d), and initializing simulations with either a low accumulation rate ($a_c = 0.1$ m/yr) that starts on the landward side of the retrograde section (panels a,c) or high accumulation rate ($a_c = 0.8$ m/yr) that start on the seaward side of the retrograde section (panels b,d). In all simulations, accumulation rate is then slowly ramped to $a_c = 0.3$ m/yr over 30 kyr (with all other parameters values held constant as listed in Tables 1 and 2). This parameter space picture captures the most significant modifications of simple thermal oscillatory behavior by retrograde bed topography. Longer or steeper retrograde sections than specified in the above ranges are not included because they may peak above sea level, conflicting with the model assumption that the bed is always below sea level at the grounding line. The parameter range spanned in Figure 5 is sufficient for mapping the behavior regimes described in the preceding sections.

Retrograde sections located far upstream of the range of grounding line migration associated with thermal oscillations in the baseline simulation ($x_0 + L < 690$ km) or of short length ($L < 80$ km) have a minimal impact on ice stream behavior. However, when the retrograde section is located near the minimum grounding line position the grounding line may persist for centuries on the retrograde section during ice stream stagnation (marked by white "x" marks in Figure 5). Such behavior occurs for a wide range of $b_{xr}$, indicating that even strong retrograde slopes cannot prevent the reversal of grounding line migration during the onset of ice stream activation. There is no evidence that the marine ice sheet instability plays any significant role for ice streams which stagnate on or near sufficiently small retrograde sections ($L < 50$ km), regardless of the steepness of the retrograde slope.

Shallow retrograde sections located around the maximum grounding line position result in prolonged intervals of several centuries where the grounding line of an active ice stream advances onto and then retreats from a retrograde section (such cases are marked by white circles in Figure 5). These long, shallow retrograde sections may also amplify thermal oscillations by forcing the grounding line to advance through the same process involved with the marine ice sheet instability. Such behavior is limited to relatively shallow retrograde slopes, indicating that the marine ice sheet instability plays a more important role here and prevents such behavior entirely for steeper retrograde slopes.

When the retrograde section is located around the maximum grounding line position and is sufficiently steep, thermal oscillations are suppressed completely. In this case, the steep retrograde section causes the grounding line to advance sufficiently quickly that ice flow changes associated with thermal oscillations are not able to reverse the direction of grounding line migration. The result is that the ice stream attains a steady state grounding line position seaward of the retrograde section.

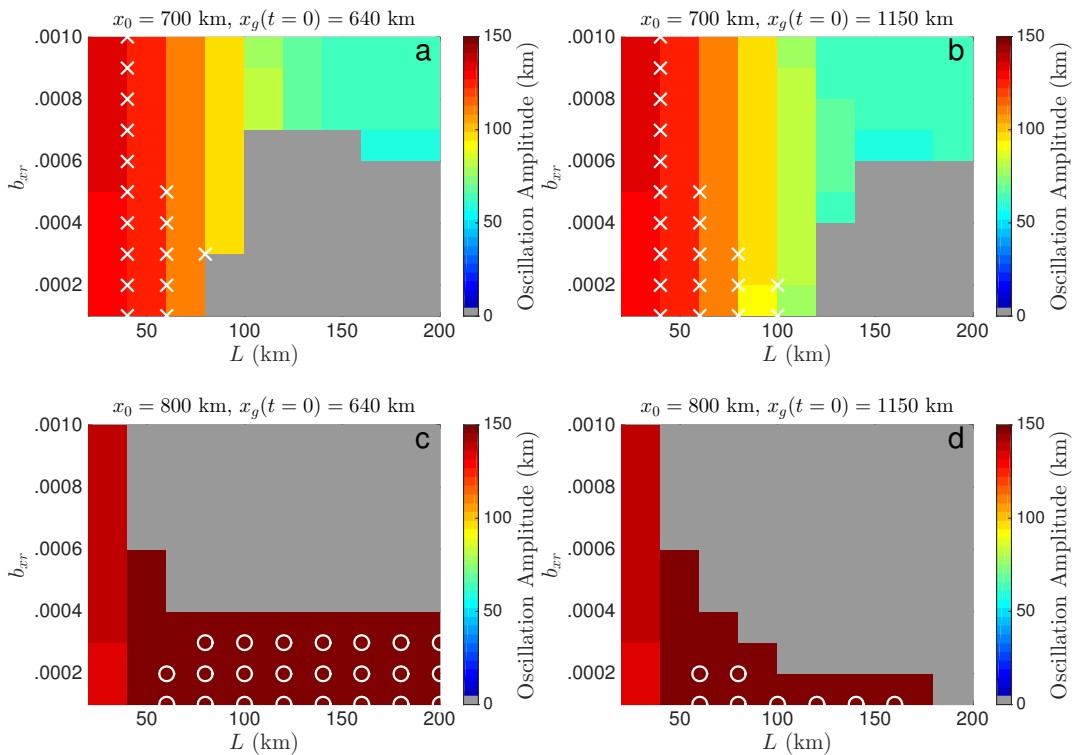

**Figure 5.** Amplitude of grounding line migration associated with thermal oscillations as a function of length ($L$) and slope ($b_{xr}$) of retrograde section. Simulations with zero oscillation amplitude are shaded in grey. "x" markers indicate simulations where the grounding line is on a retrograde slope at its minimum position during the stagnant phase. Circle markers indicate simulations where the grounding line is on a retrograde slope at its maximum position during the active phase. Panels (a) and (b) have a retrograde section starting at $x_0 = 700$ km. Panels (c) and (d) have a retrograde section starting at $x_0 = 800$ km. Panels (a) and (c) have initial ice stream grounding line position landward of retrograde section at $x_g(t = 0) = 640$ km. Panels (b) and (d) have initial ice stream grounding line position seaward of retrograde section at $x_g(t = 0) = 1150$ km. In the baseline simulation the grounding line position oscillations between 690 km and 815 km.

At even steeper slopes, thermal oscillations reappear, but are reduced in amplitude relative to the baseline simulation and are restricted to the bed downstream of the retrograde section. However, there is still a wide gap in oscillation amplitude between parameter regimes where oscillations are suppressed and those where it is reduced entirely. A small increase in slope ($b_{xr}$) of the retrograde section (e.g. along a transect where $L = 150$ km in Figure 5a,b) results in a transition from steady state to finite-amplitude oscillations in grounding line position. We have yet to find any evidence that the thermal mechanism is capable of producing oscillations of arbitrarily weak amplitude (Robel et al., 2013).

Hysteresis behavior associated with retrograde slopes is important because it imprints ice stream history on future behavior and so must be considered when spinning up numerical ice stream models. In section 3, we showed that for sufficiently steep

retrograde slopes, the grounding line either oscillates on the landward side of the retrograde slope, attains a steady state, or undergoes a limit cycle oscillation on the seaward side with, at most, small excursions onto the retrograde slope. The hysteresis shown in Figure 2c for this case is analogous to that predicted by the marine ice sheet instability for fixed bed properties. We also see evidence for such hysteresis in a narrow range of retrograde sections with intermediate $b_{xr}$ and $L$. Simulations with retrograde sections of intermediate length ($80 \lesssim L \lesssim 140$ km) and located near the minimum position of grounding line oscillations retain oscillatory behavior when initialized seaward of the retrograde section, but not always when initialized from landward. Additionally, simulations with retrograde sections of intermediate slope ($.0002 \lesssim L \lesssim .0004$) and located near the maximum position of grounding line oscillations retain oscillatory behavior when initialized landward of the retrograde section, but not always when initialized from seaward.

## 5  Conclusions

There are two main conclusions to be drawn from the results of this study. First, we have demonstrated that grounding line position and behavior change significantly as a result of spatiotemporal variations in bed properties that arise as a part of unforced ice flow variability. While ice streams which have a hard bed or a permanently-saturated soft bed may not necessarily undergo large internal variations in basal shear stress, in a broad parameter regime relevant to observed ice streams, the possibility of a dynamically-evolving soft bed must be considered. As we have shown, models which do not include dynamically-varying bed properties are not able to simulate a wide array of ice stream behaviors.

Second, ice stream grounding lines on or near retrograde slopes may exhibit behaviors which are not predicted by existing theories for grounding line stability or internal ice stream variability. Ice streams which exhibit internal variability in bed properties may persist on retrograde slopes for centuries and reverse their direction of migration on such slopes. The marine ice sheet instability is not currently equipped to explain changes in ice flux through the grounding line associated with time-dependent bed properties, which complicates how observations of grounding line migration are interpreted. The grounding line of an ice stream retreating onto a section of retrograde slope may continue to retreat irreversibly, or, may pause for centuries during stagnation before re-advancing onto the prograde slope. The latter possibility may explain why the grounding line of the currently stagnant Kamb Ice Stream rests on a retrograde slope (Fried et al., 2014) without evidence for ongoing migration (Horgan and Anandakrishnan, 2006). Of course, as we discuss below, ice shelf buttressing may alternately be the dominant process responsible for stabilizing the grounding line of Kamb Ice Stream. Determining which processes influence the stability and variability of grounding line position is critical in making accurate long-term predictions of ice stream flux and grounding line position. However, such a task is made more difficult by the short observational records of grounding line position for many ice streams, which extend (at most) only a few decades. To make more progress in contextualizing existing observations and predicting future ice stream behavior, numerical models are necessary. To admit the widest range of possible ice stream behavior, ice stream modeling studies must include dynamic variation in bed properties or convincingly argue that such variation is not applicable to a particular ice stream.

There are 3-D thermomechanical ice stream models which include more processes than the simple flowline model utilized in this study. The simplicity of this flowline model enables exploration of the parameter space, process-level understanding of dynamical behavior and comparison to theories of grounding line stability and ice stream variability. Other factors, such as strong ice shelf buttressing, are known to play an important role in grounding line stability (Goldberg et al., 2009; Katz and Worster, 2010; Gudmundsson et al., 2012), and in suppressing thermal oscillations (Robel et al., 2014). Consequently, these processes may dominate the dynamics under some circumstances. However, to the extent that oscillations in velocity and grounding line position occur in some ice streams - and observations show that they do (Retzlaff and Bentley, 1993; Catania et al., 2012) - the dynamics discussed in this study are potentially relevant to both models and observations. Though we have neglected internal strain heating from this model, simulations with strain heating included indicate that vertical and longitudinal strains of horizontal velocity contribute negligibly to the overall heat budget of the ice stream, even during periods of strong longitudinal strain, such as activation. The model used in this study is laterally integrated over the ice stream to form a longitudinal flowline, and so lateral shear heating should correspondingly be spread over the entire width of the ice stream (though shear is zero at the actual centerline). In this case, shear heating contributes less than 1% to the heat budget. However, if we instead considered a flowline through the shear margin (though this in conflict with other assumptions made in model formulation), then lateral shear heating is non-negligible (10%). Nonetheless, in order to draw definitive conclusions about specific ice streams, more complete 3-D ice stream modeling is needed, which takes into account variations in bed properties simulated in this study and also details not captured by a flowline model, such as fully dynamic ice shelf buttressing, cross-stream variations in basal topography, lateral advection and shear margin migration (Haseloff et al., 2015).

We do not challenge the notion that an unbuttressed ice stream with basal shear stress that is only a function of sliding velocity cannot have a stable steady state grounding line on a retrograde bed slope (excluding other physical processes). However, there are limitations in trying to fit all observed ice sheet dynamics to this simple version of the marine ice sheet instability. Indeed, recent studies (reviewed in the introduction) have shown that an array of potentially important processes not included in the canonical formulation of the marine ice sheet instability may inhibit it. Other studies (Schmeltz et al., 2002; Joughin et al., 2009, 2010; Parizek et al., 2013) have also shown that forced simulations of unstable grounding line migration over retrograde slopes are highly sensitive to assumptions about bed properties. The simulations presented here show the types of grounding line behavior that are possible once we step away from the limiting confines of the classical marine ice sheet instability.

A new generation of observations (Schroeder et al., 2013; Smith et al., 2013) and inverse models (Sergienko and Hindmarsh, 2013) have provided indications that subglacial hydrology may provide a stabilizing feedback in situations where there is high basal shear stress near the grounding line of an ice stream under significant external forcing, such as ocean melting. Other studies (e.g. Wolovick et al., 2014) have shown that patches of high basal traction can travel along ice streams on forcing-relevant time scales. However, the paucity of direct observations of bed conditions over most of West Antarctica, including the presence (or lack) of plastic till and meltwater and their time-dependent evolution, is still a major obstacle to making accurate predictions of ice stream evolution. Future studies must continue to find new ways to make observations of bed properties, which can be incorporated into models with dynamically evolving subglacial hydrology. Such an approach would enhance dynamical understanding of ice sheet stability and improve predictions of Antarctic Ice Sheet change.

*Acknowledgements.* All model results used to produce the figures in this study and flowline model code are available from the corresponding author upon request. This work has been supported by the NSF grant AGS-1303604 (AR and ET). ET thanks the Weizmann Institute for its hospitality during parts of this work. AR has been supported by the NSF Graduate Research Fellowship. CS has been supported by NSERC Discovery Grant 357193-12.

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
