# Peer review of "Persistence and Variability of Ice Stream Grounding Lines on Retrograde Bed Slopes"

_The Cryosphere, 2016_

## Referee Comment (RC1) · Anonymous Referee #1 · 7 May 2016

**Review of a manuscript "Persistence and Variability of Ice Stream Grounding Lines on Retrograde Bed Slopes" by A. A. Robel, C. Schoof and E. Tziperman.**

This study concerns with stability of the grounding line on beds with retrograde slopes under variable external (surface accumulation) and internal (basal) conditions. The authors use a one-dimensional flow-line model complemented with a two-dimensional (one horizontal and one vertical) ice thermodynamic model and a model of subglacial till evolution. The results show that under various parameter combinations, the grounding line can exhibit reach behaviours having a stable steady-state position, oscillating, being stable on a retrograde slope and then readvancing, etc. This study sheds light on previously disregarded aspects of a more than four-decades old theory of the marine ice-sheet instability. Overall, the manuscript is well-written and will be a substantial contribution to the existing body of literature on this subject.

**Genearl comments**

There are several questions/concerns with the model formulation and the performed experiments. The model description is very brief and lacks details about assumptions and approximations. Though the model is described in Robel et al. (2014), it wold be useful to provide a self-contained description of the model without asking a reader to refer to another publication. For instance, the authors mention basal meltwater (p. 4 line 8), however, the mass-balance equation (2) has only the accumulation rate $a_c$. It is unclear whither the basal melt rate is disregarded in this equation because it is much smaller than the accumulation rate, or because $a_c$ is the net accumulation, i.e. the difference between the surface and basal ablation/accumulation rates, or because of some other assumption. It also appears (eqn (4) in Robel et al., 2014) that the basal melting/freezing rate is disregarded in computations of the vertical velocity, $w$. It is unclear why. Even though the ice flow model accounts for the lateral shear (eqn (1)), the advection-diffusion equation for ice temperature (eqn (7) in Robel et al., 2014) does not include the internal heating due to ice

deformation. There is no explanation why the authors have chosen to disregard it.

The authors mention that the model horizontal resolution in the grounding zone is 100 m, however, there is no indication whether the results (the mode of the grounding line behaviour and its specific location) are sensitive to this value. For instance, the authors find two instances of hysteresis for a very narrow range of the accumulation rate, $a_c$, 6 mm/yr or <2% (p. 7 line 13). It would be interesting to know whether the same result holds for higher spatial resolutions, and how numerical errors compare to such changes in the model parameters.

In the Conclusions section, in addition to mentioning indirect observations of subglacial water, the authors may consider mentioning inverse modeling results indicating highly variable basal conditions in the vicinity of the grounding lines (e.g. Sergienko and Hindmarsh, 2013) and numerical modeling studies exploring the effects of traveling patches of high/low basal traction on ice flow (e.g. Wolowick et al, 2014).

**Specific Comments**

Many parameters from table 1 are not used in this manuscript.

Figure 5 is difficult to relate to the model parameters. Adding a sketch illustrating what parameters are varied in what experiments may be helpful.

**References**

Robel, A., Schoof, C., and Tziperman, E. (2014). Rapid grounding line migration induced by internal ice stream variability, J. Geophys. Res., 119, 2430–2447.

Sergienko O. V. and R. C. A. Hindmarsh. 2013. Regular patterns in frictional resistance of ice-stream beds seen by surface data inversion, Science, 342(6162), 1086-1089, doi: 10.1126/science.1243903

Wolovick, M. J., T. T. Creyts, W. R. Buck, and R. E. Bell (2014), Traveling slippery patches produce thickness-scale folds in ice sheets, Geophysical Research Letters, 41, doi:10.1002/2014GL062248.

---

## Referee Comment (RC2) · Anonymous Referee #2 · 9 May 2016

Summary

Ice streams display a wide range of behaviors, including unforced variability as well as reversal of grounding-line migration across and/or sustained stabilization on retrograde slopes. Using a thermomechanical flowline model that excludes ice-shelf buttressing in order to focus solely on the potential impacts of dynamically-varying bed properties, the authors are able to generate a wide range of ice-streaming behaviors across regions of retrograde beds that are dependent on the length of the retrograde segment, its slope, and whether the grounding line advances or retreats towards that segment. This work logically builds on previous studies by the authors and others as it illustrates a wide array of ice dynamics that are often counter to those predicted by the MISI feedback on retrograde beds, but are predicted simply by the interaction of a dynamically-evolving plastic bed with regions of retrograde slope.

[Figure]

General Comments

This is a nice, largely well-written piece of work that independently supports previous findings by once again illustrating how critical basal rheology and prior ice-flow history (assumed initial conditions) are to our ability to predict the future evolution of streaming ice flow. My main concern, as discussed below, is that the explanations for changes in simulated streaming behavior rely heavily on omitted discussions of how the model treats important boundary conditions that ultimately lead to the reported behavior. The logical progression of thoughts is thereby lost within Section 4. With a few important revisions to the text, I believe this will become an important, publishable contribution to our field.

Section 2: There is too much reliance on Robel et al., 2014. More needs to be included here for this to be a stand-alone publishable unit. For example: What is your ice-front condition for both your momentum and mass balance? These are too important to omit. I'm assuming (but shouldn't have to) that you are including a balance between water pressure and longitudinal stress at the ice front (including a sea-level line in your schematic will also visually highlight that you are simulating a marine ice-front condition). Your treatment of the flux condition at the grounding line should be stated so that discussion of advance and retreat is better framed and logical for the reader.

Subsections within 4: With the above omissions in Section 2, what should be clear and logical to the reader is often counter-intuitive in this part of the paper (see specific comments below). Because the findings and explanations within Section 4 serve as the foundation of this important contribution, Section 2 needs to be revisited by the authors.

Added discussion: With variable bed properties and oscillatory stagnation/activation of streaming flow, a discussion on the impact of omitting vertical shear on your results is warranted (unless vertical shear is indeed treated, as in Robel et al., 2014, and feeds back into the ice softness; again, not clear). With additional viscous dissipation

(does not appear in the equations in Robel et al., 2014, so I assume is not included here) and softening of the ice, I would suspect that some of the transitions in behavior might be muted due to both thermal (reduction of the thermal gradient above frozen, or nearly frozen, regions) and dynamic (softening) feedbacks. I would consider an additional simulation or two where you vary Abar both temporally and spatially over and just upstream and downstream of regions where you have pronounced gradients in basal sliding to at least address the dynamic question and then use those findings to include a brief supporting statement.

Minor Specific Comments

p2, line2: I would suggest adding the impact of pinning points: "which buttress ice sheets through lateral contact with bedrock and/or localized basal contact with bathymetric highs"

p4, Fig 1: see above comment regarding the inclusion of sea level.

p4, line5: Again, without the context of how you treat the ice flux once it reaches the grounding line, it is not clear why the grounding line must retreat (or advance). What drives a retreat? How is mass removed from the system? A short description of the ice-front condition applied when solving your continuity equation will clear all of this up.

p4, lines5-6: Although you are assuming a plastic bed (implicitly the bed strength>=basal shear stress), this sentence should be reworded in terms of till strength to be clear that you are not implying a force imbalance in the upstream direction (i.e., rather than basal shear stress + lateral shear stress > driving stress, I believe you meant to state bed strength + lateral shear stress > driving stress): "Eventually, till becomes sufficiently strong that the combined basal shear strength and lateral shear stress exceeds the driving stress and the ice stream stagnates."

p5, lines22-24: Another sentence or two on how the critical accumulation rate values remove the solution branches would be beneficial.

p6, last complete sentence: Rather than just stating that there is a lack of aperiodic oscillations, it would be informative to also offer insights into why you think there is this notable difference in behavior.

p8, line2: Here, I believe "over a wide range of parameter values" continues to refer to a wide range of a_c values, not variations in empirical parameters in your sliding law (again, not actually included in this manuscript) related to bed properties. Given how this paragraph begins, consider rewording to explicitly state: "over a wide range of a_c values."

p10, line6: Consider adding a reference to Fig. 3a here: "In contrast, when bed properties are allowed to freely vary (Figure 3a), such a retreat..."

p10, lines8-9: Again, without explaining how grounding-line migration is treated, this discussion is not intuitive. Why should grounding line retreat necessarily be a response to the accumulation of ice thickness and the deformation of ice, which could drive more ice across the grounding zone and lead to an advance? Sorry to belabor the point, but without a discussion of how you are treating key processes within the current manuscript, the understanding of temporally and spatially varying dominant processes is unnecessarily muddled. This is a solid contribution and these minor issues are easily remedied.

p11, lines7-8: Same problem... not clear why extra ice advected to the ice front, leading to thickening there, doesn't promote ice-front advance.

p11, lines8-12: And because the above is not clear, this discussion isn't intuitive (although, I am sure it would be with additional discussion of the ice-front and grounding-line treatment).

Sections 4.3 and 4.5: The explanations discussed here rely on clearing up the description of the enhanced overshoot in the previous section.

Technical Corrections

p7, line22: bifurcations remain (rather than remains)

p10, line7: This situation is contrary to what happens with MISI (it stops short of a full retreat off the retrograde bed), so shouldn't this be worded: "…such a retreat may not occur when yield stress…" rather than "may occur"?

p11, line13: delete the extra "to the"

p11, line19: "of" should be "or"

p11, line31: Consider rearranging to better prime the reader for the negative feedback to come: "…the positive feedback initially dominates, causing an overshoot…

p14, line16: parameter regimes

---

## Author Comment (AC1) · 23 May 2016

Response to Reviewer Comments 1

Review of a manuscript "Persistence and Variability of Ice Stream Grounding Lines on Retrograde Bed Slopes" by A. A. Robel, C. Schoof and E. Tziperman.

This study concerns with stability of the grounding line on beds with retrograde slopes under variable external (surface accumulation) and internal (basal) conditions. The authors use a one- dimensional flow-line model complemented with a two-dimensional (one horizontal and one vertical) ice thermodynamic model and a model of subglacial till evolution. The results show that under various parameter combinations, the grounding line can exhibit reach behaviours having a stable steady-state position, oscillating, being stable on a retrograde slope and then readvancing, etc. This study sheds light on previously disregarded aspects of a more than four-decades old theory of the marine ice-sheet instability. Overall, the manuscript is well-written and will be a substantial contribution to the existing body of literature on this subject.
We thank the reviewer for these generous comments and thoughtful suggestions on this manuscript. As we indicate below, we have added substantial descriptions of the model formulation to help frame the later discussion and save the reader from referring to Robel et al. 2014. Our detailed responses to your comments are inline below.

General comments
There are several questions/concerns with the model formulation and the performed experiments. The model description is very brief and lacks details about assumptions and approximations. Though the model is described in Robel et al. (2014), it wold be useful to provide a self-contained description of the model without asking a reader to refer to another publication. For instance, the authors mention basal meltwater (p. 4 line 8), however, the mass-balance equation (2) has only the accumulation rate ac. It is unclear whither the basal melt rate is disregarded in this equation because it is much smaller than the accumulation rate, or because ac is the net accumulation, i.e. the difference between the surface and basal ablation/accumulation rates, or because of some other assumption. It also appears (eqn (4) in Robel et al., 2014) that the basal melting/freezing rate is disregarded in computations of the vertical velocity, w. It is unclear why.
These are all good points, and we can see how these would be natural questions, which one would have to hunt for in Robel et al. (2014) to answer. We have added language which indicates that basal melt rate is neglected because it is much smaller than accumulation rates and vertical velocities in this model configuration. We have also explicitly defined how vertical velocity is calculated in this model.

Even though the ice flow model accounts for the lateral shear (eqn (1)), the advection-diffusion equation for ice temperature (eqn (7) in Robel et al., 2014) does not include the internal heating due to ice 1 deformation. There is no explanation why the authors have chosen to disregard it.

We have added a discussion of this omission (in addition to a description of how temperature is calculated). As Suckale et al. (2014) argues, deformation-induced heating is only significant in the shear margins, and so we omit it in this central flowline ice stream model.

The authors mention that the model horizontal resolution in the grounding zone is 100 m, however, there is no indication whether the results (the mode of the grounding line behaviour and its specific location) are sensitive to this value. For instance, the authors find two instances of hysteresis for a very narrow range of the accumulation rate, ac, 6 mm/yr or $< 2\%$ (p. 7 line 13). It would be interesting to know whether the same result holds for higher spatial resolutions, and how numerical errors compare to such changes in the model parameters.
We show in Robel et al. (2014) that the range of grounding line migration is converged at these horizontal resolutions. We have added an additional sentence to point this out.

In the Conclusions section, in addition to mentioning indirect observations of subglacial water, the authors may consider mentioning inverse modeling results indicating highly variable basal conditions in the vicinity of the grounding lines (e.g. Sergienko and Hindmarsh, 2013) and numerical modeling studies exploring the effects of traveling patches of high/low basal traction on ice flow (e.g. Wolowick et al, 2014).
These others studies and evidence of strong traction have been added in the conclusions.

Specific Comments
Many parameters from table 1 are not used in this manuscript.
We have removed unreferenced parameters.

Figure 5 is difficult to relate to the model parameters. Adding a sketch illustrating what parameters are varied in what experiments may be helpful.
We have added table 2, which lists all parameter variations for all simulations in this study.

**References**

Robel, A., Schoof, C., and Tziperman, E. (2014). Rapid grounding line migration induced by internal ice stream variability. *J. Geophys. Res.*, 119:2430–2447.

Suckale, J., Platt, J. D., Perol, T., and Rice, J. R. (2014). Deformation-induced melting in the margins of the west antarctic ice streams. *Journal of Geophysical Research: Earth Surface*, 119(5):1004–1025.

---

## Author Comment (AC2) · 23 May 2016

Response to Reviewer Comments 2

Summary
Ice streams display a wide range of behaviors, including unforced variability as well as reversal of grounding-line migration across and/or sustained stabilization on retrograde slopes. Using a thermomechanical flowline model that excludes ice-shelf buttressing in order to focus solely on the potential impacts of dynamically-varying bed properties, the authors are able to generate a wide range of ice-streaming behaviors across regions of retrograde beds that are dependent on the length of the retrograde segment, its slope, and whether the grounding line advances or retreats towards that segment. This work logically builds on previous studies by the authors and others as it illustrates a wide array of ice dynamics that are often counter to those predicted by the MISI feedback on retrograde beds, but are predicted simply by the interaction of a dynamically-evolving plastic bed with regions of retrograde slope.

General Comments
This is a nice, largely well-written piece of work that independently supports previous findings by once again illustrating how critical basal rheology and prior ice-flow history (assumed initial conditions) are to our ability to predict the future evolution of streaming ice flow. My main concern, as discussed below, is that the explanations for changes in simulated streaming behavior rely heavily on omitted discussions of how the model treats important boundary conditions that ultimately lead to the reported behavior. The logical progression of thoughts is thereby lost within Section 4. With a few important revisions to the text, I believe this will become an important, publishable contribution to our field.
Thank you for the very helpful comments! As you will notice, we have added much more description of the model, including important boundary conditions.

Section 2: There is too much reliance on Robel et al., 2014. More needs to be included here for this to be a stand-alone publishable unit. For example: What is your ice-front condition for both your momentum and mass balance? These are too important to omit. I'm assuming (but shouldn't have to) that you are including a balance between water pressure and longitudinal stress at the ice front (including a sea-level line in your schematic will also visually highlight that you are simulating a marine ice-front condition). Your treatment of the flux condition at the grounding line should be stated so that discussion of advance and retreat is better framed and logical for the reader.
This is a good point. We have added the stress and flotation boundary conditions at the grounding line to set up the discussion of advance and retreat. We have also discussed how flux through the grounding line is treat and added a sea level line to Figure 1.

Subsections within 4: With the above omissions in Section 2, what should be clear and logical to the reader is often counter-intuitive in this part of the paper (see specific comments below). Because the findings and explanations within Section 4 serve as the foundation of this important contribution, Section 2 needs to be revisited by the authors.
Added discussion: With variable bed properties and oscillatory stagnation/activation of streaming flow, a discussion on the impact of omitting vertical shear on your results is warranted (unless vertical shear is indeed treated, as in Robel et al., 2014, and feeds back into the ice softness; again, not clear). With additional viscous dissipation (does not appear in the equations in Robel et al., 2014, so I assume is not included here) and softening of the ice, I would suspect that some of the transitions in behavior might be muted due to both thermal (reduction of the thermal gradient above frozen, or nearly frozen, regions) and dynamic (softening) feedbacks. I would consider an additional simulation or two where you vary Abar both temporally and spatially over and just upstream and downstream of regions where you have pronounced gradients in basal sliding to at least address the dynamic question and then use those findings to include a brief supporting statement.

It was certainly unclear, without referring to Robel et al. (2014), that vertical shear in velocity and spatiotemporally-variable $\bar{A}$ are indeed included in all the simulations in this manuscript. In our expanded description of the model formulation, these points are included. Internal deformational heating is not included, as we expect it to be small in the ice stream interior (as shown in Suckale et al. 2014) and at the location of activation waves, small in comparison to frictional dissipation at the bed. This is also discussed in the expanded section on model formulation.

Minor Specific Comments
p2, line2: I would suggest adding the impact of pinning points: "which buttress ice sheets through lateral contact with bedrock and/or localized basal contact with bathymetric highs"
Added

p4, Fig 1: see above comment regarding the inclusion of sea level.
Added.

p4, line5: Again, without the context of how you treat the ice flux once it reaches the grounding line, it is not clear why the grounding line must retreat (or advance). What drives a retreat? How is mass removed from the system? A short description of the ice-front condition applied when solving your continuity equation will clear all of this up.
We have added a complete description of the stress and flotation boundary conditions at the grounding line and sentences stating that ice flux at the grounding line removes mass from the system and changes in grounding line ice thickness drive migration of the grounding line.

p4, lines5-6: Although you are assuming a plastic bed (implicitly the bed strength$\geq$basal shear

stress), this sentence should be reworded in terms of till strength to be clear that you are not implying a force imbalance in the upstream direction (i.e., rather than basal shear stress + lateral shear stress > driving stress, I believe you meant to state bed strength + lateral shear stress > driving stress): "Eventually, till becomes sufficiently strong that the combined basal shear strength and lateral shear stress exceeds the driving stress and the ice stream stagnates."
Very perceptive point, this has been changed in the way you suggested.

p5, lines22-24: Another sentence or two on how the critical accumulation rate values remove the solution branches would be beneficial.
Sentence added on the physical basis for the loss of stability on retrograde slopes.

p6, last complete sentence: Rather than just stating that there is a lack of aperiodic oscillations, it would be informative to also offer insights into why you think there is this notable difference in behavior.
We added an explanation in this sentence to indicate that the regularity of oscillations is likely due to the lack of interactions with other ice streams, which are cited as mechanisms for aperiodicity in Brinkerhoff and Johnson (2015).

p8, line2: Here, I believe "over a wide range of parameter values" continues to refer to a wide range of $a_c$ values, not variations in empirical parameters in your sliding law (again, not actually included in this manuscript) related to bed properties. Given how this paragraph begins, consider rewording to explicitly state: "over a wide range of $a_c$ values."
Done

p10, line6: Consider adding a reference to Fig. 3a here: "In contrast, when bed properties are allowed to freely vary (Figure 3a), such a retreat. . ."
Added

p10, lines8-9: Again, without explaining how grounding-line migration is treated, this discussion is not intuitive. Why should grounding line retreat necessarily be a response to the accumulation of ice thickness and the deformation of ice, which could drive more ice across the grounding zone and lead to an advance? Sorry to belabor the point, but without a discussion of how you are treating key processes within the current manuscript, the understanding of temporally and spatially varying dominant processes is unnecessarily muddled. This is a solid contribution and these minor issues are easily remedied.
This is a good comment for a critical part of the paper. As mentioned above, we have added a more detailed description of the processes that drive grounding line migration in section 2. We have also added further description throughout this entire paragraph to describe how imbalances

in ice flux at the grounding line during different stages of the thermal oscillation cycle drive this behavior.

In addition to the added description of grounding line migration in section 2, there was definitely a key confusion in how these sentence were written. In the early part of the active phase, the grounding line undergoes thickening as ice is delivered from upstream, but thinning occurs upstream as the reservoir of ice is depleted by high velocities. Then at the end of the active phase, deactivation begins at the ice divide, causing reduces ice flux from upstream (and thinning), but then reaches the grounding line, causing reduced grounding line flux (and thickening). These confusions have been cleared up by adding additional sentences here and taking more time to discuss these processes step-by-step.

Sections 4.3 and 4.5: The explanations discussed here rely on clearing up the description of the enhanced overshoot in the previous section.
We have cleared up the description of enhanced "overshoot", and added additional explicit references to this "overshoot" starting in section 2 to clarify that this same process is relevant to explaining each of these different cases.

Technical Corrections
p7, line22: bifurcations remain (rather than remains)
Fixed

p10, line7: This situation is contrary to what happens with MISI (it stops short of a full retreat off the retrograde bed), so shouldn't this be worded: ". . .such a retreat may not occur when yield stress. . ." rather than "may occur"?
Good point, we have changed it to this

p11, line13: delete the extra "to the"
Fixed

p11, line19: "of" should be "or"
Fixed

p11, line31: Consider rearranging to better prime the reader for the negative feedback to come: ". . .the positive feedback initially dominates, causing an overshoot. . ."
Fixed

p14, line16: parameter regimes
Fixed

**References**

Brinkerhoff, D. and Johnson, J. (2015). Dynamics of thermally induced ice streams simulated with a higher-order flow model. *Journal of Geophysical Research: Earth Surface*, 120(9):1743–1770.

Robel, A., Schoof, C., and Tziperman, E. (2014). Rapid grounding line migration induced by internal ice stream variability. *J. Geophys. Res.*, 119:2430–2447.

---

## Referee Report (RR1)

**Review of a manuscript "Persistence and Variability of Ice Stream Grounding Lines on Retrograde Bed Slopes" by A. A. Robel, C. Schoof and E. Tziperman.**

This is a revised version of a previously submitted manuscript. The authors have addressed most of the major reviewers comments, however, there are a few left. Though, these issues are not crucial, and do not preclude the manuscript publication, their clarification would certainly benefit the manuscript.

The first issue is the negligence of the internal heating due to ice deformation. The authors refer to a study by Suckale et al. (2014) that found that this term *is* (not "may be") important in shear margins. A model used by Suckale et al. (2014) assumes that ice thickness is constant across an ice stream and that the vertical velocity, which determines the vertical advection, is scaled linearly with the surface accumulation. Both assumptions are violated in the shear margins. For the present study, it would be more straightforward to perform an extra simulation that includes the internal heating term and compare it with a simulation without that term. Presumably, the model runs are fairly fast. A quantitative estimate of the effect of internal heating on the model behaviour would make the presented results more robust.

The second issue is extremely narrow range of the accumulation rate, 6 mm/yr, that results in the hystereses (p. 9, line 14). It is likely, that inter-annual variability of the surface accumulation rate of the present-day ice streams is larger than this value. Although, It is difficult to prove it, as both, the observational errors of the current accumulation rate, and instrumental and methodological errors of the past accumulation rates inferred from radar observations are substantially larger than this very narrow range of the critical accumulation rates. The manuscript would benefit from some discussion what are physical implications of such a narrow range and how realistic the obtained hysteresis behaviour.

**Minor comments**

Abstract: line 4 "numerical" is unnecessary.

Introduction: p.1 lines 11-12: The first two sentences are unnecessary.

Eqn (1): should be $\rho_i$ instead of $\rho$ and it should be defined after this equation.

Model preliminaries: p.3 line 26: "Vertical shear of horizontal velocity is assumed to arise independently..." is (a) unclear (arise independently of what?), and (b) does not sound right - vertical shear is deformation, which is determined by the vertical structure of horizontal velocity. But it is not a property or characteristic of velocity.

P. 4, line 2: what does "$x - z$ mass continuity" mean? Simply saying that $w$ is determined by the vertical integration of the mass-continuity equation would remove ambiguities.

P. 4, line 8: no need to repeat that the melt rate is neglected. It is already stated on line 4.

P. 5, line 16: what does "self-consistently" mean? What are "activation waves"?

---

## Author Response (AR3)

Review of a manuscript "Persistence and Variability of Ice Stream Grounding Lines on Retrograde Bed Slopes" by A. A. Robel, C. Schoof and E. Tziperman. This is a revised version of a previously submitted manuscript. The authors have addressed most of the major reviewers comments, however, there are a few left. Though, these issues are not crucial, and do not preclude the manuscript publication, their clarification would certainly benefit the manuscript.

Thank you for this thorough review. You will find below that we have made all these modifications (and learned some things along the way about strain heating!).

The first issue is the negligence of the internal heating due to ice deformation. The authors refer to a study by Suckale et al. (2014) that found that this term is (not "may be") important in shear margins. A model used by Suckale et al. (2014) assumes that ice thickness is constant across an ice stream and that the vertical velocity, which determines the vertical advection, is scaled linearly with the surface accumulation. Both assumptions are violated in the shear margins. For the present study, it would be more straightforward to perform an extra simulation that includes the internal heating term and compare it with a simulation without that term. Presumably, the model runs are fairly fast. A quantitative estimate of the effect of internal heating on the model behaviour would make the presented results more robust.

We have run these additional simulations. What we found is that the $u_x$ and $u_z$ strain heating terms are pretty much always negligible, even during periods of activation. Lateral shear heating $(u_y)$ pretty much only occurs in the shear margins, and is negligible (order 1%) when spread over the entire width of the ice stream (to be consistent with the rest of the laterally-integrated model formulation). However, just to be thorough we have also conducted simulations where the flowline is imagined to be through the shear margin (hence the lateral strain heating is not spread out so much and is approximately 10x the previous version). In this case, lateral shear heating can be important (as other have found for shear margins - not surprising). We have added all of this discussion to the conclusions section which is now referenced in the model section. The reference to Suckale et al. has been removed in accordance with your suggestion.

The second issue is extremely narrow range of the accumulation rate, 6 mm/yr, that results in the hystereses (p. 9, line 14). It is likely, that inter-annual variability of the surface accumulation rate of the present-day ice streams is larger than this value. Although, It is difficult to prove it, as both, the observational errors of the current accumulation rate, and instrumental and methodological errors of the past accumulation rates inferred from radar observations are substantially larger than this very narrow range of the critical accumulation rates. The manuscript would benefit from some discussion what are physical implications of such a narrow range and how realistic the obtained hysteresis behaviour.

We have added a few sentences of discussion indicating this exact issue with narrow parameter regimes and urging the use of perturbed-physics ensembles in complex ice sheet model studies

so as not to run single simulations which fall in a narrow parameter regime.

Minor comments
Abstract: line 4 "numerical" is unnecessary.
Fixed
Introduction: p.1 lines 11-12: The first two sentences are unnecessary.
Removed
Eqn (1): should be $\rho_i$ instead of $\rho$ and it should be defined after this equation.
Fixed
Model preliminaries: p.3 line 26: ?Vertical shear of horizontal velocity is assumed to arise independently. . . ? is (a) unclear (arise independently of what?), and (b) does not sound right - vertical shear is deformation, which is determined by the vertical structure of horizontal velocity. But it is not a property or characteristic of velocity.
Fixed
P. 4, line 2: what does "x-z mass continuity" mean? Simply saying that w is determined by the vertical integration of the mass-continuity equation would remove ambiguities.
Rewritten
P. 4, line 8: no need to repeat that the melt rate is neglected. It is already stated on line 4.
Removed
P. 5, line 16: what does "self-consistently" mean? What are ?activation waves??
Re-written

[revised manuscript text omitted]

---

## Author Response (AR4)

Dear Dr. Robel and co-authors, Thank you for re-submitting your revised manuscript and responding in full to the second review. I am now almost ready to accept your manuscript. In my final readthrough, I found the paper, in general, very well written (and certainly very interesting) but picked up a handful of minor grammatical issues, appended in the attached. I would also ask you to consider the use of hyphens in, for example, "ice stream", "ice sheet" and "grounding line" whenever these terms are used as prepositions, i.e. one can discuss the case of an ice stream (no hyphen) versus ice-stream processes. I corrected these cases in the abstract (and the paper title!), but request that you use a find/replace exercise to correct this through the manuscript. Apologies for these final, very minor issues, but I am sure you will them very simple to implement. When you're done, just follow the instructions for resubmitting appended to this message, and I should be in position to accept the manuscript.

Thank you to the editor for this final very thorough review. You will find that we have made all the modifications as you have asked.